**METHOD**                                                                                     **Open Access**

# diffBUM-HMM: a robust statistical modeling approach for detecting RNA flexibility changes in high-throughput structure probing data

Paolo Marangio[2,3][†] (iD), Ka Ying Toby Law[1†] (iD), Guido Sanguinetti[1,2,3]* (iD) and Sander Granneman[1]* (iD)

*Correspondence:
gsanguin@sissa.it;
sander.granneman@ed.ac.uk
[†]Paolo Marangio and Ka Ying Toby
Law contributed equally to this
work.
[1]Centre for Synthetic and Systems
Biology, The University of
Edinburgh, Edinburgh, UK
[3]SISSA Data Science Excellence
Department Initiative, Trieste, Italy
Full list of author information is
available at the end of the article

## Abstract

Advancing RNA structural probing techniques with next-generation sequencing has generated demands for complementary computational tools to robustly extract RNA structural information amidst sampling noise and variability. We present diffBUM-HMM, a noise-aware model that enables accurate detection of RNA flexibility and conformational changes from high-throughput RNA structure-probing data. diffBUM-HMM is widely compatible, accounting for sampling variation and sequence coverage biases, and displays higher sensitivity than existing methods while robust against false positives. Our analyses of datasets generated with a variety of RNA probing chemistries demonstrate the value of diffBUM-HMM for quantitatively detecting RNA structural changes and RNA-binding protein binding sites.

**Keywords:** Hidden Markov model, High-throughput RNA structure probing, RNA structural changes

## Background

Understanding the structure of RNA is key to unravel its in vivo function, and it is also highly relevant to biomedicine, drug discovery, and synthetic biology [1–4]. Recent years have witnessed a blossoming of high-throughput methods that couple next-generation sequencing with biochemical assays to 'probe' the structure of thousands of RNA molecules simultaneously, including whole transcriptomes [5–16]. The majority of these biochemical assays use reagents such as SHAPE (selective 2′-hydroxyl acylation analyzed by primer extension) reagents [5–7, 16–20] and dimethyl sulfate (DMS) [8, 10, 21]. These chemicals modify the 2′-hydroxyl (OH) group of riboses or bases of flexible/single-stranded nucleotides, respectively, and the sites of modification can be detected by performing a reverse transcription (RT) reaction. A major advantage of using chemical probes is that some are also highly effective for probing RNA structure in liv-

ing cells [16, 18, 22–24], making it possible to compare in vivo and in vitro structures, and reveal potential protein-binding sites [16, 22]. Depending on the RT enzyme and the reaction chemistry used, the modification either causes the RT enzyme to terminate transcription, resulting in truncated cDNAs, or to skip the adduct, frequently introducing mutations (SHAPE-MaP; [5, 21]). Following on from this, the site and degree of nucleotide modification can be extracted from NGS data by quantifying how frequently the RT terminated at a given nucleotide position [6, 8–10] or by calculating mutation frequencies for each nucleotide [5, 21]. Although NGS has a number of unprecedented advantages in terms of sensitivity and the number of molecules that can be analyzed simultaneously, the analysis of the resulting data is not trivial and exhibits significant challenges. Depending on the cDNA library preparation method used, biases in sequence representation and read coverage can be introduced [25], and there can also be quite significant inter-replicate variability in untreated (control) and treated samples [26]. To specifically address these issues, we recently developed a probabilistic modeling pipeline called beta-uniform mixture hidden Markov model (BUM-HMM) [27]. One of the strengths of BUM-HMM is that it analyzes the inter-replicate variability of samples in the treatment and control pools. Moreover, it adopts an empirical statistical analysis method that obviates the need of conventional data correction and normalization techniques that are used in the majority of the analysis pipelines. Although BUM-HMM generates statistically sound estimates of nucleotide accessibility at the nucleotide level, its probabilistic output does not represent an absolute value that quantifies the degree of accessibility of RNA at a particular nucleotide. Therefore, it is not immediately usable for differential analyses between different treatments.

The ability to accurately detect nucleotide regions that differentially react with RNA structure probing reagents under diverse conditions, or due to the effect of mutations, is of great importance to researchers. As a consequence, the last few years have seen an increase in the development of a number of bioinformatics tools to detect differentially reactive nucleotides (DRNs) in RNA structure probing datasets. Amongst the available tools are classSNitch [28], PARCEL [29], RASA [30], deltaSHAPE [22], StrucDiff [31], and the recently published dStruct [32]. In particular, dStruct has been shown to perform best by recording the lowest false-positive rate, while offering compatibility with a wide range of existing RNA structure probing datasets. However, one possible limitation of dStruct is that the pipeline uses a variety of statistical tests to predict DRNs. As a result, dStruct corrects for multiple hypothesis testing, which likely makes it conservative with its predictions. Hence, we reasoned that a method that does not rely on statistical tests but rather on a model and posterior probability, such as BUM-HMM, would be preferable, because it would be inherently less vulnerable to problems associated with multiple hypothesis testing. In addition, dStruct uses SHAPE reactivity values as input, which involves normalization and outlier elimination strategies on quantitative data to generate a reactivity profile for each nucleotide. Since the distribution of quantitative data often differs between probing experiments, such data normalization procedures might result in useful data being removed. In contrast, the BUM-HMM model uses only the raw counts for each nucleotide (i.e. read coverage and either total RT drop-offs or mutation counts). It also employs empirical statistical analyses that preserves the independent distribution of each dataset while being robust to outliers. To test whether the BUM-HMM algorithm

could be useful for detecting DRNs, we extended the model to develop diffBUM-HMM (differential BUM-HMM). We used diffBUM-HMM to compare a number of publicly available RNA structure probing datasets and benchmarked the tool against dStruct [32]. Similar to dStruct, diffBUM-HMM effectively identified DRNs in the datasets; however, consistent with our hypothesis, it exhibited higher sensitivity and, like dStruct, has a very low false-positive rate. Because diffBUM-HMM is compatible with a wide variety of high-throughput RNA structure probing methods, it should be of general interest to the RNA community.

## Results

### diffBUM-HMM Model

diffBUM-HMM is a natural extension of BUM-HMM (Fig. 1). An intermediate step of BUM-HMM is the computation of an empirical *P* value for each treatment-control comparison at each nucleotide position. Each empirical *P* value is then passed onto a hidden Markov model. BUM-HMM has a *hidden state* $h_t$ ($t$ = 1, 2, 3, …, $T$ for $T$ nucleotides) representing the true binary state of the $t$th nucleotide (M = modified by the probe; U = unmodified by the probe) and the *observed variable* $v_t$, which is the empirical *P* value at that position. For diffBUM-HMM, the *hidden state* is expanded to take on four potential values instead of two: nucleotide is unmodified in both conditions (UU; *hidden state* 1); nucleotide is unmodified in the 1st condition but modified in the 2nd (UM; *hidden state* 2); nucleotide is modified in the 1st condition, but unmodified in the 2nd (MU; *hidden state* 3); nucleotide is modified in both conditions (MM; *hidden state* 4). In turn, the *observed variable* at each state is now represented by two *P* values rather than one. As

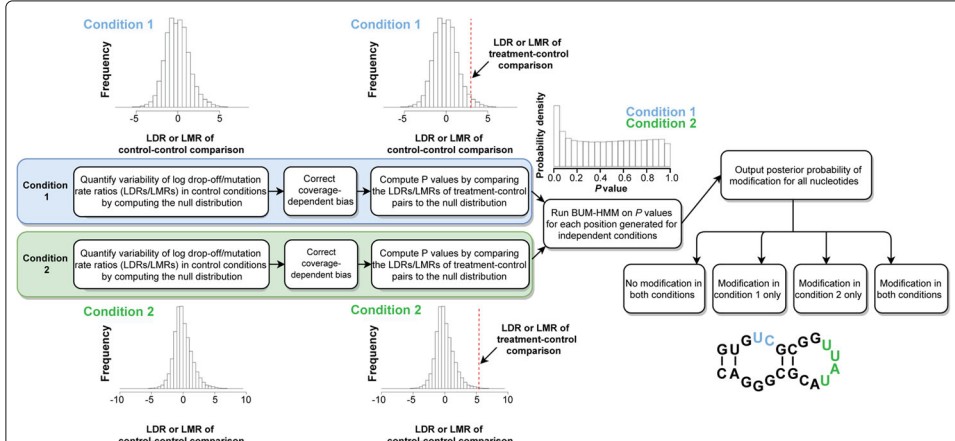

**Fig. 1** Overview of the diffBUM-HMM computational analysis pipeline. For each experimental condition (e.g. conditions 1 and 2), the log-ratios of drop-off/mutation rates (LDRs/LMRs) at each nucleotide position are computed for pairs of control samples to give a null distribution, in order to quantify variability in drop-off or mutation rates observed by chance. LDRs/LMRs are also computed similarly for all possible treatment-control comparisons. Coverage-dependent biases are then removed by applying a variance stabilization transformation. Subsequently, per-nucleotide empirical *P* values are computed for all possible treatment-control comparisons in each condition, by comparing the corresponding log-ratios to the null distribution. diffBUM-HMM is run on *P* values associated with the two independent conditions as observations, leaving out any nucleotides with missing data. The resulting output is a posterior probability of modification for each nucleotide, ranging from 0 to 1. diffBUM-HMM reports whether nucleotides were unmodified in both conditions, modified in either of the conditions or modified in both conditions

the *hidden state* can take on four possible values, extending BUM-HMM to diffBUM-HMM entails increasing the size of the transition matrix from $2 \times 2$ to $4 \times 4$ and adapting its values. While in principle the expectation-maximization algorithm could be used to identify directly this transition matrix from data, we found that adapting the original BUM-HMM heuristic values to diffBUM-HMM by assuming independence of the two conditions yielded good results. A sensitivity analysis confirmed the validity of this approach (Additional file 1: Figure S1).

### diffBUM-HMM prediction of structural changes in the 35S pre-rRNA of yeast ribosome synthesis mutants

To test diffBUM-HMM, we first reanalyzed the high-throughput structure probing datasets generated from two mutant *Saccharomyces cerevisiae* strains that express structurally distinct pre-ribosomal RNA (pre-rRNA) precursors [33]. These ChemModSeq-type [6] high-throughput datasets were selected because (a) the read coverage for the pre-rRNAs analyzed was very high (i.e. >10,000 reads per nucleotide) and (b) some of the regions that were predicted to be structurally distinct based on sequencing results have been verified by primer extension (PE) analysis. Since PE analysis is still considered to be one of the most reliable biochemical approaches for detecting sites of chemical modification, we used the PE data as 'ground truth' for evaluating the goodness of the DRNs predicted by the tools benchmarked in this study, including diffBUM-HMM, deltaSHAPE, and dStruct. Our analysis of these ChemModSeq data revealed that diffBUM-HMM and deltaSHAPE reported a large number of DRNs, while dStruct reported 32 differentially reactive regions (DRRs) throughout the length of the 35S pre-rRNA when using a 5-nucleotide search length and a false discovery rate (FDR) cutoff of 0.15 (Fig. 2A).

In our previous study [33], we validated some of the ChemModSeq results by performing PE analysis on several regions in the 5′ external transcribed spacer (5′ ETS) as well as the 5′ end of 18S (Fig. 2B, regions highlighted in gray; Fig. 3). Here, the ChemModSeq analyses predicted a high concentration of DRNs, which were largely confirmed by the PE data (Fig. 3). deltaSHAPE and diffBUM-HMM identified many nucleotides as DRNs that also showed differential reactivity in the PE data (Fig. 3A, B). In the 5′ ETS, the patterns of the ChemModSeq SHAPE reactivity profiles and the regions verified by PE were very similar (Fig. 3A, B), suggesting that the ChemModSeq high-throughput data for the 5′ ETS is of high quality. Both deltaSHAPE and diffBUM-HMM identified DRNs in these regions of the 5′ ETS that also appeared differentially modified in the PE data. These regions reportedly contain two U3 snoRNA base-pairing sites (281–291 and 464–479), which plays an essential role in the processing of the 35S pre-rRNA and the maturation of the 90S pre-ribosome. These structure probing data analyses showed that deleting the fifth RNA-binding domain in Mrd1 (Mrd1∆5 mutation) significantly decreased SHAPE reactivities in these U3 snoRNA base-pairing sites. This implied that in this Mrd1 mutant, the U3 snoRNA remains base-paired to the 5′ ETS, and we concluded that Mrd1-dependent remodeling of pre-ribosomes is required for the timely release of the U3 snoRNA from these large complexes [33]. More specifically, the PE data showed that nucleotides 282 and 283 in the 35S pre-rRNA were more reactive in the strain expressing the wild-type Mrd1 protein compared to the Mrd1∆5 mutant. In the wild-type Mrd1 strain, nucleotides 470–479 in the 5′ ETS were noticeably more reactive compared to the ∆5 mutant. These

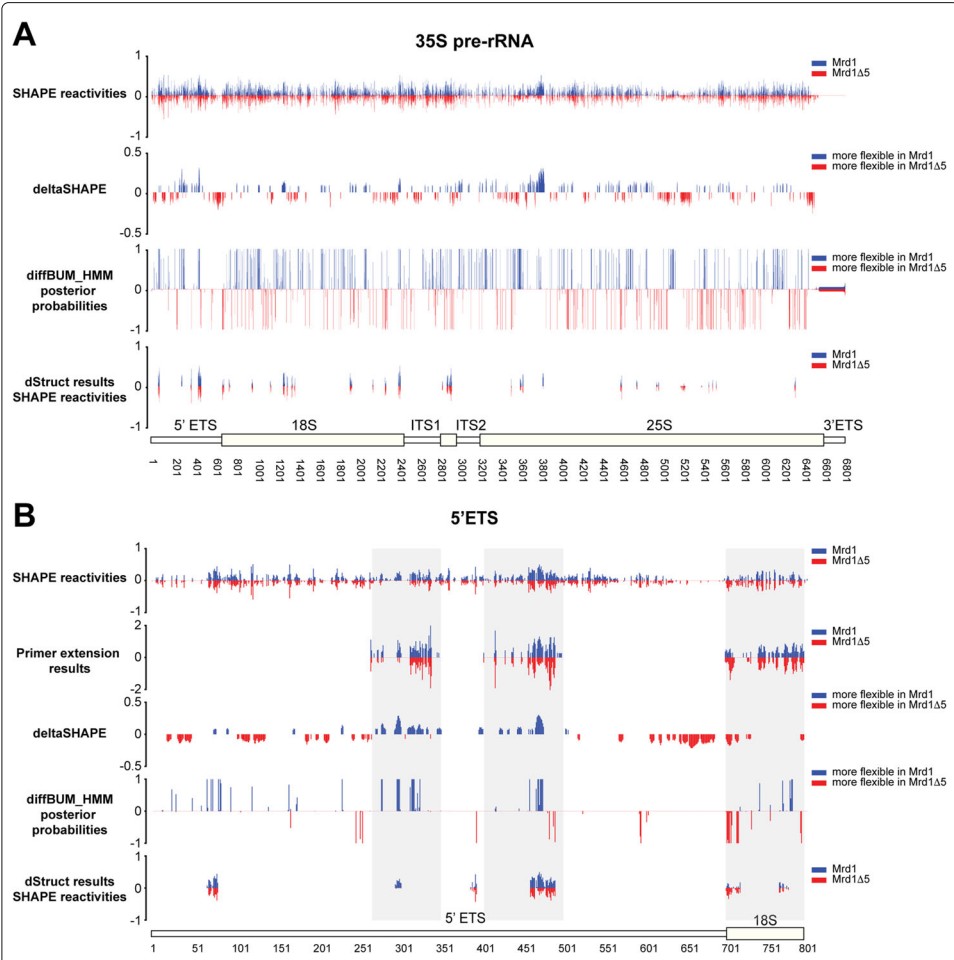

**Fig. 2** diffBUM-HMM effectively detects differentially reactive nucleotides in the earliest detectable yeast pre-rRNA precursor. **A** The top panel shows the SHAPE reactivities [33] from the first biological replicate of both wild-type Mrd1 and Mrd1Δ5 deletion mutant, which were used to identify DRNs with deltaSHAPE. The deltaSHAPE values were calculated according to [22]. For the deltaSHAPE panel, positive values indicate the position of nucleotides that are more reactive in pre-rRNA associated with wild-type Mrd1, whereas negative values indicate the position of nucleotides that are more reactive in pre-rRNA associated with the Mrd1Δ5 mutant. The same data was reanalyzed using the diffBUM-HMM and dStruct algorithms (panels 4 and 5, respectively). For the diffBUM-HMM results, the posterior probabilities for differential states were calculated using the raw counts. For the dStruct analyses, 2–8% normalized RT drop-off rates were used, as recommended by the authors [32]. **B** The same as in **A** but only for the 5′ ETS and 5′ end of 18S rRNA. We additionally included the results from the PE analysis (panel 2). The gray areas indicate the regions validated by PE analysis

PE results closely corresponded with the diffBUM-HMM output. When using the recommended 5-nucleotide search window, dStruct reported 4 DRRs in the 5′ ETS (Fig. 3). Two DRRs (298–306, 463–493) coincided with the regions that were indicated to be differentially reactive by PE (268–352 and 405–502). The latter DRR overlaps with one of the U3 snoRNA base pairing sites, but the length of the reported DRR is quite extensive, without clear indication of which nucleotides are differentially reactive between the two samples (Fig. 3B). The DRRs reported by dStruct were also reserved to regions that have more considerable differences in reactivity, while diffBUM-HMM also identified nucleotides with more modest but highly reproducible differences in SHAPE reactivity.

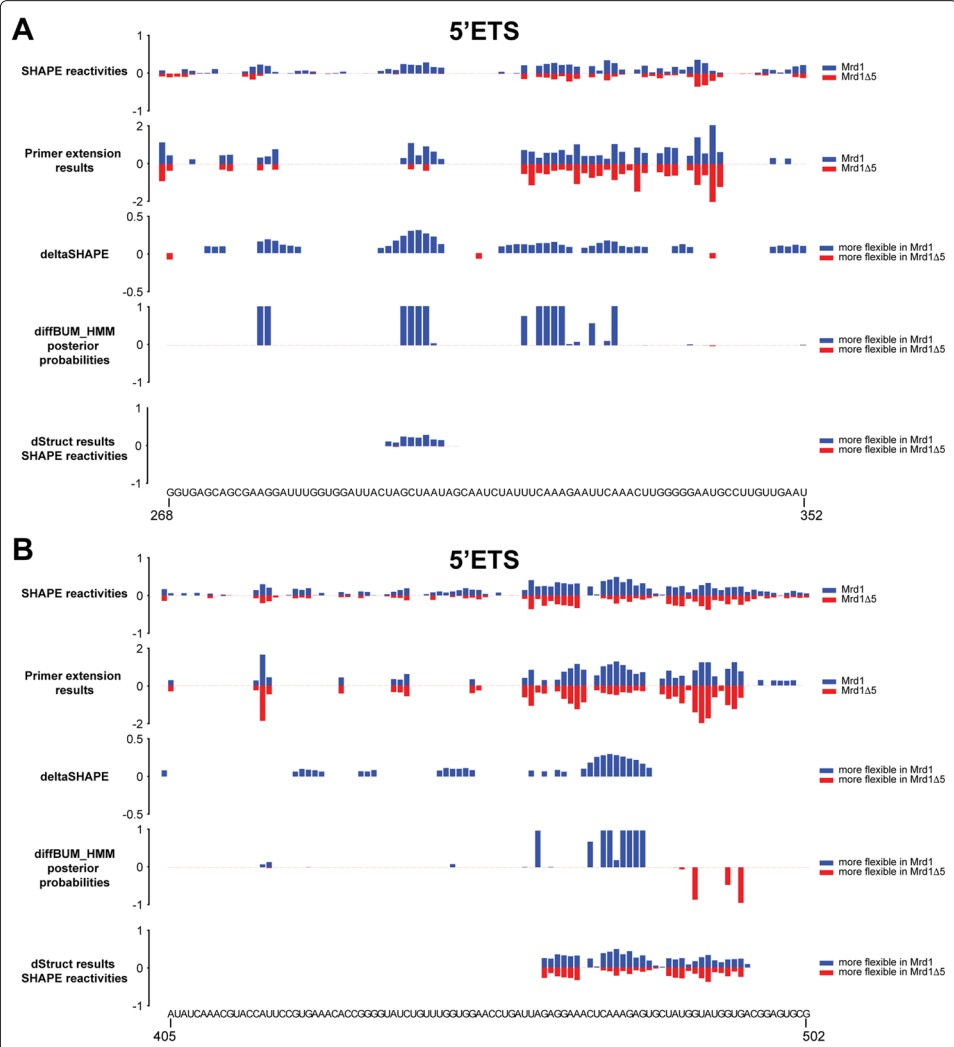

**Fig. 3** diffBUM-HMM detects differentially reactive nucleotides (DRNs) in the 5′ ETS of the 35S pre-rRNA precursor. **A**, **B** SHAPE reactivities, deltaSHAPE, diffBUM-HMM, and dStruct analysis results for two regions (positions 268–352 and 405–502) within the 5′ ETS. The top panel shows the SHAPE reactivities [33] from the first biological replicate, which were used to identify DRNs with deltaSHAPE (deltaSHAPE panel). Positive values indicate the 1M7 nucleotide reactivities in pre-rRNA associated with wild-type Mrd1, whereas negative values indicate the reactivities in pre-rRNA associated with the Mrd1 deletion (Δ5) mutant. The second panel shows the quantification of the PE analysis for these regions. The same data was reanalyzed using the diffBUM-HMM and dStruct algorithms (panels 3 and 4, respectively). For the diffBUM-HMM results, the posterior probabilites for differential reactivity were calculated using the raw counts. For the dStruct analyses, 2–8% normalized RT drop-off rates were used, as recommended by the authors [32]

dStruct only searches for DRRs that are longer than a user-specified threshold and lower than a predefined FDR, and the outcome of the results was strongly influenced by what settings were used for these parameters. For example, when we used an 11-nucleotide search window, dStruct reported 3 DRRs with an FDR of ≤0.05. However, these DRRs were quite extensive (23–60 nucleotides). With the aim of increasing the resolution of the dStruct results, we also repeated the analyses using a 1-nucleotide search length. However, this did not result in any DRRs with an FDR of ≤0.15. We conclude that, compared

to the current gold standard dStruct, diffBUM-HMM detects DRNs with much higher sensitivity and resolution.

### diffBUM-HMM calls no false positives in datasets generated from identically treated RNA samples

Despite the fact that deltaSHAPE and diffBUM-HMM were able to detect more experimentally verified DRNs in the 35S dataset, it is plausible that this apparent higher sensitivity is, at least in part, the result of the low specificity of the methods. To test this possibility, we were looking for ways to calculate false-positive rates for the diffBUM-HMM algorithm. As the number of nucleotides in the 35S dataset that were verified by PE were too low to perform a meaningful analysis of false-positive rates, we reanalyzed published in vivo *S. cerevisiae* DMS Structure-Seq [32] and ChemModSeq datasets. The DMS Structure-Seq data was previously used to assess the false-positive rates of all the currently available methods for identifying DRNs [32] (see Table 1). These datasets contained biological replicates of DMS-modified and unmodified mature rRNA samples that were treated identically. Hence, the expectation would be that there would not be any DRNs detected between replicates. We re-analyzed the raw data generated from this experiment and generated drop-off rates for each nucleotide position in the four rRNAs (18S, 25S, 5S, and 5.8S). Previously, it was shown that dStruct only called three false-positive nucleotides in the DMS Structure-Seq rRNA data, whereas deltaSHAPE reported a total of 97 false positives (Table 1; [32]). Strikingly, for all the datasets analyzed, diffBUM-HMM did not report any nucleotide with posterior probability of differential modification higher than 0.4 (Fig. 4A), suggesting that diffBUM-HMM did not call any spurious DRNs. dStruct and deltaSHAPE reported 0 and 16 false positives, respectively, in the 18S rRNA ChemModSeq datasets (Table 1). DMS preferentially modifies A's and C's in flexible and single-stranded regions. Indeed, many of the 18S nucleotides called modified by diffBUM-HMM in these datasets were A's and C's that were located in single-stranded regions in the 18S secondary structure (Fig. 4B, C, also see Figure 2 in [27]). Therefore, we conclude that these DMS RNA structure probing datasets are of good quality and that diffBUM-HMM has a high specificity that is on par with dStruct and RASA.

**Table 1** Comparison of the capabilities of existing methods designed to detect differential reactive nucleotides in rRNA molecules

| Tool | Reference | Inter-replicate variability? | Noise considered? | Detection level? | False positives: 18S rRNA | False positives: 18S rRNA dataset 2 |
|------|-----------|------------------------------|-------------------|------------------|---------------------------|-------------------------------------|
| classSNitch | [28] | ✗ | ✓ | Regional | N/A | N/A |
| PARCEL | [29] | ✓ | ✗ | Regional | 61 | N/A |
| RASA | [30] | ✓ | ✓ | Regional | 4 | N/A |
| deltaSHAPE | [22] | ✗ | ✓ | Regional | 97 | 16 |
| StructDiff | [31] | ✗ | ✓ | Regional | N/A | N/A |
| dStruct | [32] | ✓ | ✓ | Regional | 3 | 0 |
| diffBUM-HMM | This work | ✓ | ✓ | Regional and Nucleotide | 0 | 0 |

The table shows the previously published results of the analyses on the identically DMS-treated yeast rRNA datasets [27, 32] as well as the results from our diffBUM-HMM analysis of these datasets. The column displaying the number of false positives indicates the number of nucleotides that were called differentially modified in DMS chemical probing data

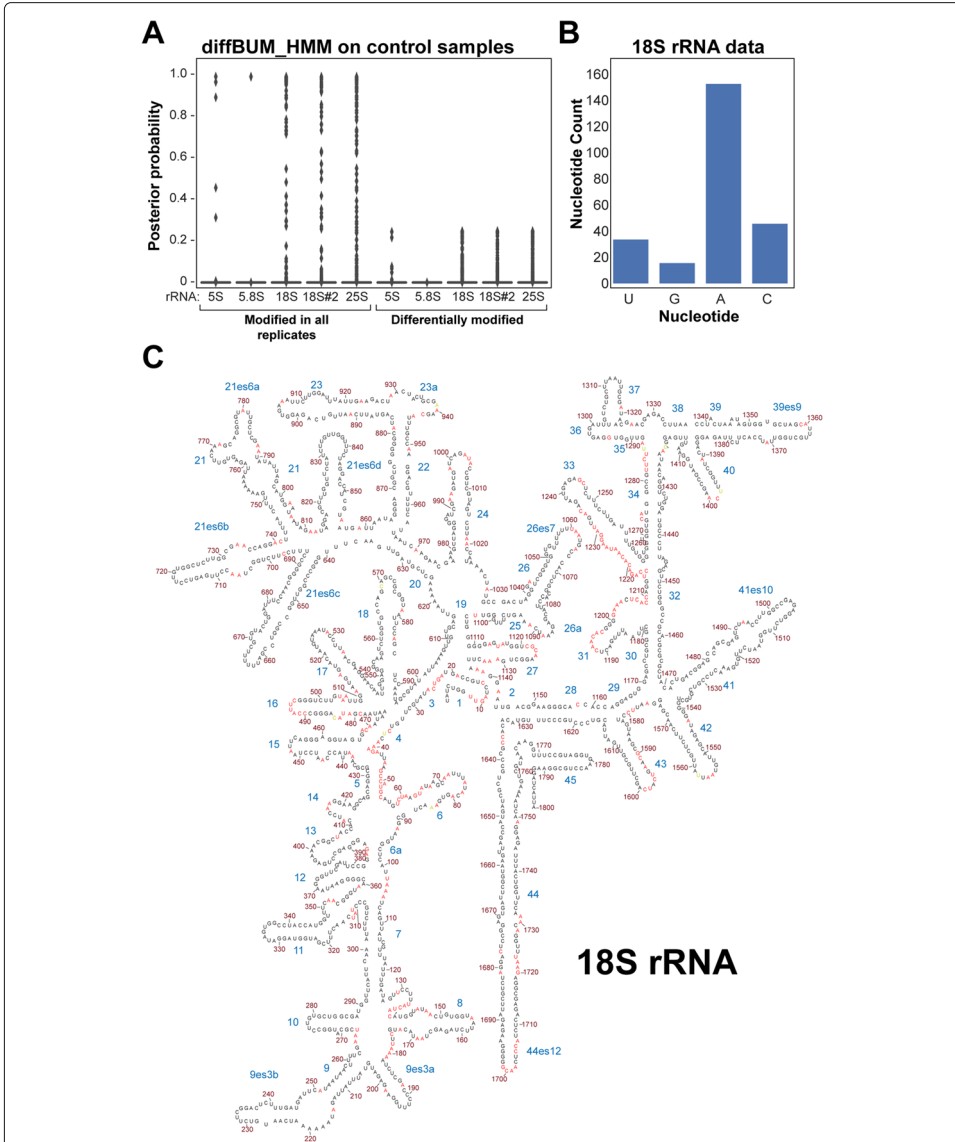

**Fig. 4** diffBUM-HMM has a very high specificity. **A** diffBUM-HMM only reports nucleotides with posterior probabilities of less than 0.4 on identically treated DMS-probed *S. cerevisiae* rRNA Structure-Seq (5.8S, 18S, 25S) and ChemModSeq 18S rRNA (18S#2) datasets. The box plot shows the distribution of the posterior probabilities for each rRNA sample. Shown are the posterior probabilities that the nucleotides were called modified in all replicates or differentially modified between replicates. **B** Base composition of nucleotides called modified in all replicates of the yeast 18S rRNA Structure-Seq data, when considering only nucleotides with posterior probabilities ≥0.95. **C** Nucleotides called modified in all replicates of the 18S rRNA Structure-Seq data (posterior probabilities ≥0.95) are highlighted in red in the secondary structure of the molecule. The names of the helices in the structure are indicated in blue

## diffBUM-HMM analysis of differentially probed Xist lncRNA

The earliest studies that reported high-throughput RNA structure chemical probing analyses relied on the reverse transcriptase falling off the modified RNA once the enzyme encountered a chemically modified nucleotide [6, 8–10, 14, 15]. However, by changing the conditions for the RT reaction, one can force a reverse transcriptase to incorporate

non-complementary nucleotides or introduce deletions into the cDNA transcript instead [5]. This approach, referred to as SHAPE-MaP (selective 2′-hydroxyl acylation analyzed by primer extension and mutational profiling), maps sites of chemical modification by analyzing the mutation frequencies of the nucleotides. To calculate SHAPE reactivities, sequencing data generated from untreated (or solvent-treated) RNA and chemically modified denatured RNA are often included. However, it has been suggested that such controls may not be essential for accurately predicting RNA structures [21, 34]. Since SHAPE-MaP essentially relies on counting the number of mutations and diffBUM-HMM relies on count data, we asked whether diffBUM-HMM can accurately detect sites of modification from SHAPE-MaP data. To test this, we reanalyzed the mouse Xist SHAPE-MaP datasets [23]. The 18-kb Xist lncRNA is essential for X-chromosome inactivation during the development of female eutherian mammals [35]. Although previous studies have suggested the importance of RNA structures in specific regions of Xist, the locations and structures of functional domains within Xist are still not well-defined. To identify Xist RNA structural features as well as regions occupied by proteins, the Weeks lab has previously performed a comprehensive SHAPE-MaP analysis of the Xist RNA that was probed in living cells (in cell/in vivo) and in protein-free (ex vivo) conditions [23]. Analyses of these data identified 33 regions in Xist that formed well-defined structures as well as many regions that could be occupied by RNA-binding proteins (RBPs). Importantly, this dataset contained two biological replicates for each condition for SHAPE-treated, untreated, and SHAPE-treated denatured RNA samples. As it was unclear whether including the denatured data in our calculations was essential, we performed the diffBUM-HMM analysis with and without normalizing the data to the mutation rates of the denatured RNA samples. An overview of the results is shown in Fig. 5. To compare our data to the deltaSHAPE results, we applied the deltaSHAPE algorithm to the individual replicates (Fig. 5A). When reactivities from the denatured data were not considered, diffBUM-HMM detected 1164 DRNs in the ex vivo condition and 188 in the in vivo condition (Fig. 5A, C). Interestingly, diffBUM-HMM reported a much larger number of DRNs in the ex vivo data than in the in vivo data relative to deltaSHAPE ($\approx$ 9-fold difference with diffBUM-HMM and $\approx$ 1.4-fold with deltaSHAPE, as shown in Fig. 5A, C). Why diffBUM-HMM calls many more DRNs in the ex vivo data is unclear; however, intuitively, one would expect that removing proteins from a very large ribonucleoprotein (RNP) complex will substantially increase the flexibility of the RNA. This is because when the RNP complex is deproteinized, those nucleotide positions normally bound by RBPs may become more accessible and/or more flexible and therefore react more readily with chemical probes. In this dataset, dStruct was also very conservative with its predictions: with a search length of 5 nt, dStruct reported 29 DRRs with a FDR of $\leq$0.15 (Fig. 5A) that range from 10 to 40 nt in size. The tool did not report regions with relatively modest but reproducible differences in reactivity, such as the region 12,000–14,000, which is enriched with FUS protein-binding sites amongst others. Remarkably, normalizing the data to the denatured samples further increased the number of DRNs detected by diffBUM-HMM and dStruct (Fig. 5A, C), with dStruct now detecting more DRRs in the 8000–9000 and 3′ regions of Xist (Fig. 5A). This confirms that including data from SHAPE-treated denatured samples can improve the detection of DRNs in SHAPE-MaP data.

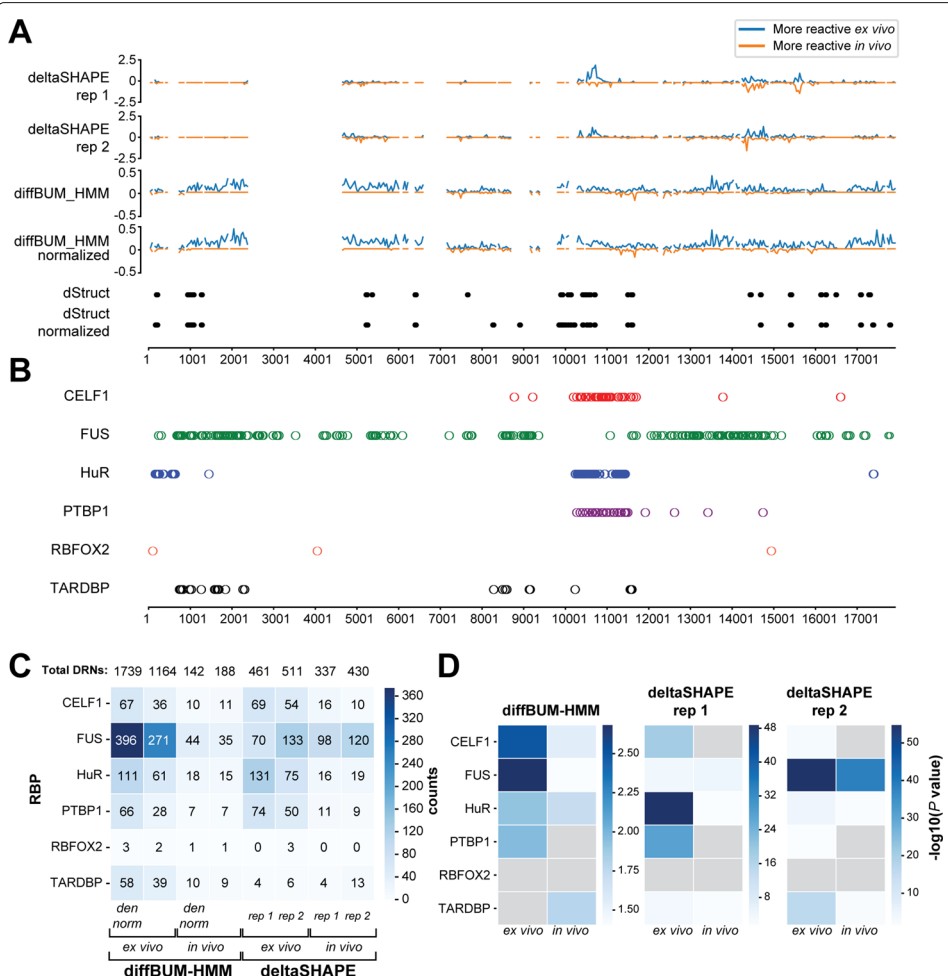

**Fig. 5** diffBUM-HMM detects a larger number of differentially modified nucleotides in the ex vivo Xist lncRNA data compared to deltaSHAPE and dStruct. **A** The differential reactivities of two deltaSHAPE replicate experiments [23] compared to the diffBUM-HMM posterior probabilities and dStruct outputs, which show the SHAPE reactivities of the regions it calls DRRs. The Xist RNA transcript was binned into 500 nucleotides regions and the differential reactivities for each bin is plotted. Regions with negative reactivities or posterior probability values are more reactive in vivo. Only those nucleotides that according to the deltaSHAPE analyses had sufficient coverage are plotted. The normalized diffBUM-HMM posteriors and dStruct panels indicate those nucleotides that were called differentially modified after normalizing the mutation rates in treated and untreated samples based on the denatured data (denoted as "den norm" in **D**). **B** Overview of the RNA-binding sites detected in the Xist transcript, as shown in [23]. **C** Overview of the number of DRNs that overlap with RNA-binding protein (RBP) binding sites in Xist in the in vivo and ex vivo data. Total DRNs indicates the total number of DRNs identified by diffBUM-HMM and deltaSHAPE in the datasets. "Den norm" indicates the data where we normalized the mutation frequencies of treated and control samples based on the denatured RNA data. **D** Enrichment of DRNs in RBP binding sites in Xist obtained from the CLIPdb database. Statistical significance for enrichment was determined using a hypergeometric test. Color legend indicates significance level, with binding sites for RBPs that are not statistically significant colored in gray

## DRNs detected in Xist using diffBUM-HMM are primarily single-stranded and enriched in protein-binding sites

A key question that we wished to address was whether the large number of additional and unique DRNs detected by diffBUM-HMM in the Xist ex vivo data were biologically meaningful. Despite the high specificity of diffBUM-HMM, we could not rule out the possibility that diffBUM-HMM simply called many false positives in this SHAPE-MaP dataset.

Deproteinizing an RNP should make sites normally occupied by RBPs more accessible to chemical probes. Therefore, we first asked whether the diffBUM-HMM DRNs were located in protein-binding sites previously identified by UV cross-linking or RNA immunoprecipitation (CLIP/RIP) experiments. The CLIPdb database contains Xist binding sites for a large number of RBPs, including CELF1, PTBP1, HuR, TARDBP, FUS, and RBFOX2 (Fig. 5B). Similar to what was previously observed in the Xist deltaSHAPE analysis [23], many of the DRNs in the ex vivo data detected by diffBUM-HMM overlapped with RNA-binding sites of these RBPs (Fig. 5B, C). When compared to the deltaSHAPE data, diffBUM-HMM identified more DRNs overlapping with FUS and TARDBP RNA-binding sites in the ex vivo data, whereas the number of ex vivo DRNs overlapping with other RBPs was comparable between the two datasets (Fig. 5C). This is presumably because most of the deltaSHAPE signal concentrated around 2–3 regions within the Xist RNA, whereas diffBUM-HMM detected DRNs throughout the transcript (Fig. 5A). We also found that in the ex vivo data for both diffBUM-HMM and deltaSHAPE many of the RBP binding sites were statistically significantly enriched for DRNs, with diffBUM-HMM DRNs preferentially enriched in CELF1 and FUS binding sites (Fig. 5D). However, diffBUM-HMM also detected many DRNs outside of these RBP binding sites, which may explain why the -log($P$ values) for binding site enrichment are overall lower compared to deltaSHAPE. This is not necessarily surprising since many other proteins bind Xist in vivo [36], and therefore, diffBUM-HMM could also be picking up binding sites from other proteins in addition to the ones reported in the CLIPdb database. As a second measure for determining whether these unique DRNs could be biologically meaningful, we performed a motif search analysis to assess whether enriched sequence motifs could be detected in regions containing DRNs. For this purpose, we grouped together DRNs located within 5 nt from each other into genomic intervals, extended these to 30 nt and analyzed sequence motif enrichment using MEME [37]. MEME detected three highly enriched motifs in the CLIPdb binding sites for CELF1, HuR and PTBP1 (Additional file 1: Figure S2). Interestingly, similar motifs could also be detected in the diffBUM-HMM and deltaSHAPE data. In the in vivo data, only a motif resembling the CELF1 binding site was significantly enriched. However, in the ex vivo data, sequences resembling HuR and PTBP1 binding sites could be detected. Moreover, diffBUM-HMM again recovered a CELF1-like motif as well as another sequence motif that was not detected in the deltaSHAPE analysis. Thus, these data strongly suggest that the DRNs dectected by diffBUM-HMM are frequently located in or near protein-binding sites.

One possible explanation for why deltaSHAPE calls fewer DRNs in the ex vivo data is because it looks within 5 nucleotide windows and only calls a given nucleotide as DRN if at least three nucleotides within that window fit the required criteria. diffBUM-HMM also assumes that DRNs are present in up to 5 nucleotide stretches; however, the algorithm will call single nucleotide DRNs if it is very clear from the data that only a single nucleotide was differentially modified. Indeed, we found that deltaSHAPE preferentially reports three nucleotide stretches, whereas diffBUM-HMM also frequently reports single nucleotide DRNs (Fig. 6A). If the DRNs uniquely detected by diffBUM-HMM indeed represent real changes in RNA flexibility, one would expect that many of these would be A's or U's as these are more frequently located in single-stranded regions such as loops or bulges. This was indeed the case (Fig. 6B). We observed the same trend in the data normalized to the data from the denatured RNA control as well as for those DRNs that

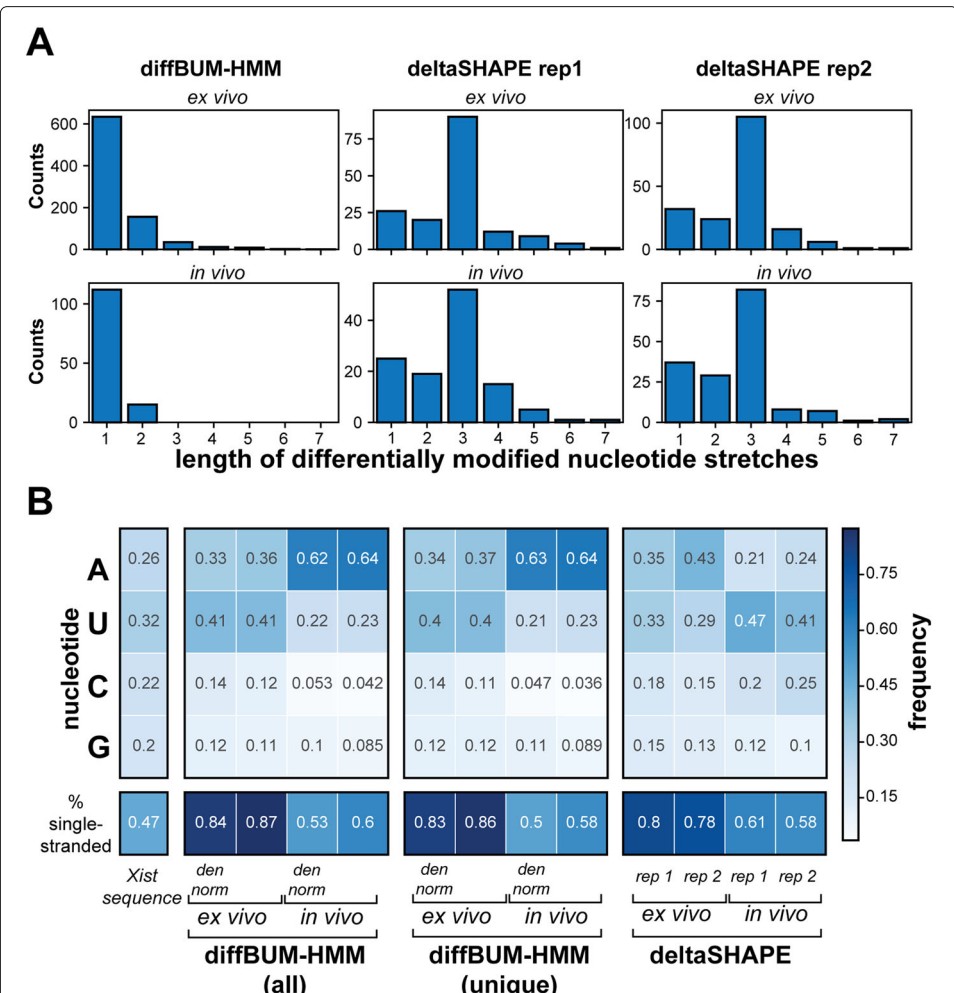

**Fig. 6** diffBUM-HMM detects more differentially reactive nucleotides (DRNs) in the Xist lncRNA that are preferentially single-stranded A's and U's. **A** diffBUM-HMM calls more single nucleotide stretches as DRNs. The barplots show the distribution of the length of stretches of nucleotides that were called DRNs by diffBUM-HMM and deltaSHAPE in the in vivo data and ex vivo data. **B** The comparison between all the DRNs called by diffBUM-HMM, including the data normalized to the denatured data, those uniquely detected by diffBUM-HMM, and the results from the deltaSHAPE analyses on the two replicates individually. diffBUM-HMM DRNs are mostly A's and U's and enriched in regions predicted to be single-stranded in Xist. DRNs identified by diffBUM-HMM are preferentially located in Xist single-stranded regions. "Den norm" indicates the data where we normalized the mutation frequencies of treated and control samples based on the denatured RNA data. "diffBUM-HMM (unique)" indicates those DRNs that were uniquely detected by diffBUM-HMM

were uniquely called by diffBUM-HMM. The deltaSHAPE results were slightly more variable but still showed a modest nucleotide preference. Because the SHAPE reagents used to modify Xist preferentially react with nucleotides in single-stranded or flexible regions, the DRNs called by diffBUM-HMM, including those uniquely detected by the tool, should be primarily located in regions that are predicted to be single-stranded in Xist. Indeed, over 80% of all the DRNs called by diffBUM-HMM in the deproteinized data were located in single-stranded regions (Fig. 6B). A few examples showing DRNs in Xist secondary structures is shown in Fig. 7. In those cases where deltaSHAPE results between replicate samples did not agree, diffBUM-HMM frequently calls the nucleotide unmodified in both conditions. However, as evident from the figures, many of the DRNs reported by

diffBUM-HMM were not detected by deltaSHAPE. Collectively, these data suggest that the DRNs detected by diffBUM-HMM in Xist represent *bona fide* changes in nucleotide flexibility that in many cases are located in single-stranded regions and overlap with or are located near protein-binding sites. In conclusion, all the available data strongly suggest that diffBUM-HMM outperforms deltaSHAPE and dStruct in both sensitivity and/or specificity.

## Discussion

### diffBUM-HMM exhibits promising advantages and functionality compared to existing methods

Over the past several years, there has been an explosion in the number of methodologies that make it possible to analyze RNA structure both in vivo and in vitro. However, the analysis of the resulting data is notoriously difficult. To be able to extract all the relevant information from the high-throughput sequencing data, many variables need to be taken into consideration. This include sequence coverage, biological variability between experiments (i.e. noise), background signal observed in untreated samples, and sequence representation bias introduced during the preparation of NGS libraries. Adding to the complexity, research groups have now started focusing on the analysis of RNA structural changes introduced by SNPs or the absence of protein binding, etc. This therefore prompted a number of labs to develop bioinformatics tools that would enable users to detect differences in RNA flexibility by comparing datasets generated under different conditions (see Table 1 for examples and references). The Aviran Lab recently published a thorough review of the pros and cons of the various methods and tested them on a variety of datasets [32], so we will not discuss this in detail here. However, that study showed that dStruct was the best performing approach, particularly when it comes to specificity. One of the great strengths of dStruct is that it is compatible with a wide variety of RNA structure probing methodologies and takes into consideration biological variability. However, as outlined above, dStruct uses a variety of statistical tests to predict DRNs within a certain sequence window. The correction for multiple hypothesis testing that dStruct employs likely also makes the tool conservative with its predictions. Indeed, our analysis of rRNA and mouse Xist SHAPE-MaP data showed that dStruct generally calls few DRRs. This prompted us to develop a tool that was based on a probabilistic graphical model as this should be less vulnerable to problems associated with multiple hypothesis testing. Here, we demonstrate that our approach (diffBUM-HMM) is indeed much more sensitive in calling DRNs compared to dStruct on all the datasets tested. However, this high sensitivity does not mean that diffBUM-HMM compromises on specificity: like dStruct, diffBUM-HMM has a very low false-positive rate. In fact, our analysis on identically treated rRNA samples probed with DMS (including rRNAs up to $\approx$ 3400 nucleotides long; [32]) revealed that diffBUM-HMM did not call any false positives.

One of the challenges we faced was the lack of datasets that would enable us to perform a more quantitative comparison between diffBUM-HMM and dStruct and other tools. The problem we have (as well as the rest of the field) is that we currently do not have a good 'ground truth' dataset that would enable us to do a meaningful statistical analysis. Although many high-throughput RNA structure probing datasets are now available, the number of datasets that describe analysis of RNA structures under different conditions

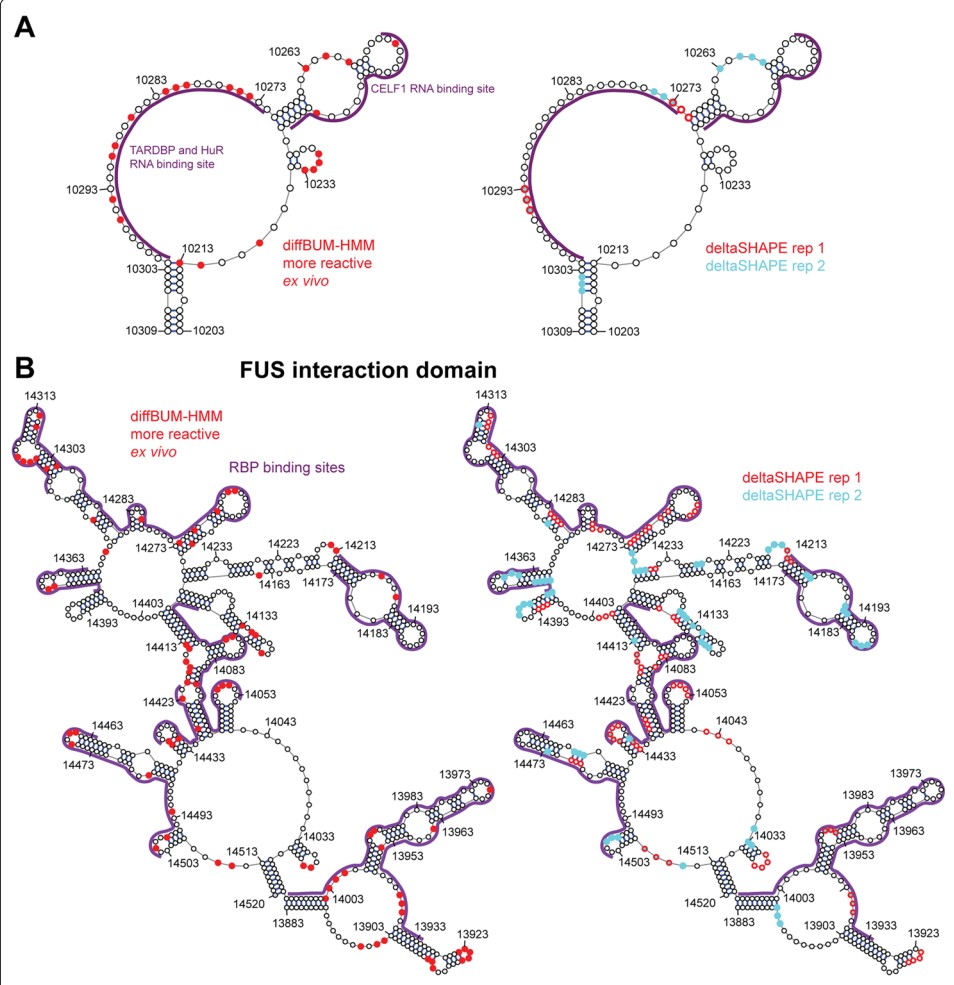

**Fig. 7** Differentially reactive nucleotides detected by diffBUM-HMM are preferentially localized in single-stranded regions and at the base of stems. **A** A secondary structure for a region in the Xist lncRNA containing CELF1, TARDBP, and HuR binding sites. The red dots indicate the nucleotides called modified only in the ex vivo data by diffBUM-HMM, while the violet lines indicate binding sites for RNA-binding proteins that were identified by CLIP/RIP (left). Also shown is the same secondary structure with the deltaSHAPE results from the two replicates individually (right). Those nucleotides called differentially modified in replicates 1 and 2 are colored red and cyan, respectively. **B** The same as in **A** but now for the FUS interaction domain of Xist

is still very limited. Additionally, diffBUM-HMM requires replicates (the more the merrier) for both treated and untreated samples, which are not readily available. For example, DMS-MaP protocols do not consider untreated control samples as the background signals that are generated under the library preparation conditions seem to be mostly stochastic and therefore not very useful [21]. To be able to perform a meaningful quantitative assessment of the available tools, we would need to have SHAPE-Seq or SHAPE-MaP data from a large RNP or macromolecular complex that was analyzed under two different conditions, and for which high-resolution structures were also generated under the exact same conditions. Unfortunately, such datasets were not yet available at the time of writing.

Our reanalysis of the Xist data revealed that diffBUM-HMM also called many more DRNs in the ex vivo data compared to deltaSHAPE and dStruct, many of which were

uniquely detected in the diffBUM-HMM analyses. We believe that the majority of these represent *bona fide* changes in RNA flexibility: SHAPE reagents preferentially react with single-stranded or flexible nucleotides, and over 80% of the DRNs detected in the deproteinized data were in regions that are single-stranded in the Xist secondary structure model (Figs. 6 and 7). Moreover, motif analyses revealed that these DRNs were also enriched in sequence motifs recognized by RNA-binding proteins (Additional file 1: Figure S2). Our analyses as well as the original Xist SHAPE-MaP paper [23] nicely illustrate how comparing in vivo and ex vivo conditions can not only help with the detection of differences in RNA structure, but also the identification of potential protein-binding sites. The observation that specific RNA-binding motifs could be detected in the deltaSHAPE, and diffBUM-HMM DRNs analyses suggests that it should even be possible to use such data to predict where on the RNA certain sequence-specific RBPs bind.

### Interpreting the output of diffBUM-HMM

Although the available evidence suggests that diffBUM-HMM is currently the best performing method for detecting DRNs, it does have a few drawbacks: the method provides a posterior probability for differential modification, which does not inform about how large the difference in chemical reactivity was between the two samples (i.e. no absolute measure of modification). To get a general impression of reactivity changes, one could inspect the drop-off rates (DORs) for the different conditions or variants alongside the output of diffBUM-HMM. For example, those regions in the 5′ ETS that reproducibly showed large changes in DORs between the wild-type and mutant Mrd1 samples (Additional file 1: Figure S6A) also had high diffBUM-HMM posterior probabilities (e.g., nt 470–480).

There could still be diffBUM-HMM predictions that at first glance may not seem to be consistent with the DORs. Two examples are nucleotides 462 and 474 in the 5′ ETS region of 35S pre-rRNA (Fig. 3B and Additional file 1: Figure S6A). In such cases, it is important to inspect the $P$ values generated by diffBUM-HMM since these values informs us on variability in the modified samples and how different they were to the control samples (Additional file 1: Figure S6B). The beta-uniform model tends to assign a higher likelihood of modification to $P$ values of $\approx 0.2$ or smaller. Looking at position 462 in the 5′ ETS region, three of the four $P$ values in the wild-type Mrd1 data are lower compared to the values found in the Mrd1 mutant data, indicating higher reactivity in the pre-rRNA samples isolated from the strain expressing the wild-type Mrd1 (Additional file 1: Figure S6B). As for nucleotide position 474, it is predicted to be most likely modified in both variants (Additional file 1: Figure S6A, fourth panel from the top, probability $\approx 0.8$), while the probability of differential modification is insignificant ($\approx 0.19$). The $P$ values for both variants are all small enough to be assigned a high likelihood of modification, but looking more closely there is also a discernible difference between the two different molecules, where the $P$ values for the wild-type Mrd1 are in a higher quantile than Mrd1$\Delta$5 (Additional file 1: Figure S6B). However, the advantage of diffBUM-HMM is that it does not only report positions that have differential activity, it also reports whether nucleotides from the two samples are modified in both conditons or not significantly reactive to the chemical. If a nucleotide is clearly modified in both conditions, albeit at different levels, it tends to predict modification in both variants/conditions. Ultimately, it is the comparative magnitude, as well as the inter-replicate consistency of the $P$ values within and across conditions

that pushes the algorithm towards a specific decision, so considering the aforementioned pieces of information in tandem would be useful when understanding the diffBUM-HMM output in this and subsequent studies.

In certain instances, there could be discrepancies between diffBUM-HMM and other measurements of differential reactivity, like deltaSHAPE. This could be because diffBUM-HMM accounts for inter-replicate variability on two different levels: the tool uses control replicates to account for background noise, and it also compares replicate samples, before proceeding to analyze for reactivity differences. On the other hand, deltaSHAPE only represents the differential reactivity detected between a single pair of samples, and does not account for inter-replicate variability with real data. Thus, the prediction of the diffBUM-HMM DRNs is based on a more comprehensive set of experimental data than a typical deltaSHAPE analysis. Hence, discrepancies in results generated by these tools could arise from variability between the biological replicates that were regarded as experimental noise by diffBUM-HMM. For example, there were two regions in the 35S pre-rRNA ChemModSeq data where the two tools did not agree (3750–3950 and 6300–6500; Fig. 2A). However, the deltaSHAPE results for each replicate were also very different in these regions, demonstrating that chemical probing data for these regions is noisy (Additional file 1: Figure S7). Similarly, there were also several regions in the Xist RNA molecule where deltaSHAPE only called differential nucleotides in one of the replicates (e.g., FUS interaction domain in Fig. 7B), and in many cases, diffBUM-HMM gave these regions low posterior probabilities of differential modification.

### Recommended input for the diffBUM-HMM algorithm

As diffBUM-HMM solely relies on nucleotide count data, it is compatible with a wide variety of high-throughput RNA structure probing methods that either measure RT drop-off or mutations (SHAPE-MaP). However, it is important to point out that diffBUM-HMM will only work well with structure probing libraries that are paired-end sequenced, as in order to quantify and correct for local variability in coverage, the precise start and end position of each cDNA in the library needs to be determined [6, 26, 27]. In our analyses, we therefore only consider reads that are properly paired (i.e. the forward and the reverse read are mapped within a specified distance on the same chromosome). Hence, diffBUM-HMM will not generate reliable results with RNA structure probing methods that rely on single-end sequencing. Paired-end sequencing is also recommended for SHAPE-MaP analysis as it would enable the selection of high-confidence mutations.

### Recent advancements in detecting RNA structural heterogeneity

diffBUM-HMM provides a nucleotide-level measure of differential accessibility; however, to obtain insights into global changes in structure, suitably constrained RNA-folding algorithms need to be used. An alternative approach called DREEM was recently proposed [34], which instead relies on a priori selecting a set of plausible structures based on the chemical probe reactivity profiles, and then determines relative shifts in abundance of the different structures via a read-clustering approach. Hence, DREEM and diffBUM-HMM perform different tasks but provide complementary information from structure

probing data sets; due to this, a direct comparison in performance between the two methods is not straightforward nor necessarily meaningful.

## Conclusions

We describe a novel modeling approach (diffBUM-HMM) for detecting changes in RNA flexibility from high-throughput RNA structure probing datasets. Our results show that diffBUM-HMM exhibits a higher sensitivity compared to the current gold standard dStruct as well as deltaSHAPE and calls very few false positives. We envision that diffBUM-HMM will be very useful for a variety of analytical tasks that pertain to different domains ranging from biomedical science to molecular genomics. diffBUM-HMM could be used to predict novel RNA regulatory elements, or study the effects of mutations on RNA structure, to pinpoint crucial functional domains in RNA or to identify novel, potential protein-binding sites within RNA. The knowledge from these studies can then be potentially applied to synthetic biology, such as the design and screen for regulators that will allow fine-tuning of arbitrary functions in synthetic gene circuits.

## Methods

### Analysis of the ChemModSeq dataset

Drop-off and read counts were generated using the pyCRAC package (https://git.ecdf. ed.ac.uk/sgrannem/pycrac) and the CRAC_pipeline_PE pipeline (https://git.ecdf.ed.ac. uk/sgrannem/crac_pipelines). Briefly, Flexbar (version 3.4.0) was used to remove adapter sequences and subsequently the reads were collapsed (pyFastqDuplicateRemover.py) to remove putative PCR duplicates. PyReadCounters from the pyCRAC package was used to calculate drop-off counts and coverage for each nucleotide position in the yeast pre-ribosomal RNAs (pre-rRNAs). These were subsequently fed to diffBUM-HMM.

### DiffBUM-HMM model

Differential BUM-HMM (diffBUM-HMM) is a variant of the beta-uniform mixture hidden Markov model (BUM-HMM) [27], and most of the modeling assumptions made for BUM-HMM also hold for diffBUM-HMM. For example, the transition probabilities are defined based on single- and double-stranded nucleotide stretches derived empirically to be of length 5 and 20, respectively. Emission probabilities follow a beta-uniform mixture model. This design is based on the expectation that nucleotides that are not modified under a given condition are associated with *P* values that follow a uniform distribution [38]. On the other hand, accessible nucleotides are associated with *P* values that follow a beta distribution, as they would exhibit LDR or LMR values that are greater than most values in the null distribution. In practice, adherence to this assumption can be easily monitored by plotting empirical *P* value distributions as in Additional file 1: Figure S3-S5. It should be pointed out that, occasionally, saturation phenomena might result in the presence of two beta peaks; for example, Additional file 1: Figure S3 shows a peak of *P* values near zero, corresponding to nucleotides which have significantly higher drop-off/mutation rates in treatment (and hence are likely modified), as well as an additional peak near 1. This peak is likely the result of saturation in the treated sample, resulting in abnormally few drop-off reads in unmodified nucleotides; the BUM-HMM likelihood will in any case assign a very low probability of modification to such

nucleotides, effectively eliminating any problem that might arise from this mismatch of hypotheses.

The $\alpha$ and $\beta$ parameters of the beta distribution were chosen heuristically to be 1 and 10, respectively. This allows to assign approximately equal likelihood under both $P$ value distribution hypotheses to nucleotides that have LDR/LMR falling in the highest quantiles of the empirical distribution. Like for BUM-HMM, it is possible to optimise the parameters of the beta-uniform distribution of diffBUM-HMM for a dataset using the expectation-maximization (EM) algorithm and Newton's optimization method. Although we provide an implementation of the EM algorithm in diffBUM-HMM, we did not use it to generate the results of this manuscript, as manual optimization already yielded very good results. The hidden Markov model takes as input continuous regions of nucleotides that satisfy a user-specified coverage threshold (i.e. non-negative threshold for all the experiments in this manuscript) and non-zero LDR/LMR for at least one treatment-control comparison. The novel aspect of diffBUM-HMM is that inference is performed based on two independent observed $P$ values, each representing a different condition. The forward-backward algorithm is the inference method for computing the posterior marginals of all hidden states.

### Analysis of enriched sequence motifs in regions containing DRNs

The Multiple EM for Motif Elicitation (MEME) tool searches for novel, recurring, and untapped motifs in given sequences [37]. To detect enriched RBP binding motifs in Xist, DRNs in the ex vivo data that were located within a window of 5 nucleotides were grouped in to a single interval, each of which was subsequently extended to 30 nucleotides using the pyNormalizeIntervalLenghts.py script from the pyCRAC package [39]. FASTA files containing the Xist sequences associated with these intervals were analyzed by MEME using the following bash command: `meme-chip -meme-minw 4 -meme-maxw 10 -meme-nmotifs 20 -meme-p 8 -meme-mod anr -norc -rna -noecho -oc OUTFILE INFILE`.

## Supplementary Information

---

**Additional file 1:**  Figure S1–S7. **Figure S1**: Optimization of the diffBUM-HMM transition matrix: boxplots of prediction mismatch value over transition matrix perturbations for the 35S and Xist molecules. **Figure S2**: Enriched RBP binding motif search amongst the DRNs/DRRs detected by diffBUM-HMM and deltaSHAPE within the Xist molecule. **Figure S3**: Distribution of log drop-off rate ratios (LDRs) and $P$ values for the 35S data. **Figure S4**: Distribution of log mutation rate ratios (LMRs) and $P$ values for the Xist data. **Figure S5**: Distribution of log drop-off rate ratios (LDRs) and $P$ values for the rRNA control datasets. **Figure S6**: Interpreting the output of diffBUM-HMM: diffBUM-HMM pipeline output in contrast with drop-off rates and $P$ values. **Figure S7**: Discrepancies between deltaSHAPE and diffBUM-HMM explained by noise in the data: A comparison between the individual replicates of deltaSHAPE analyses for the 35S molecule.

**Additional file 2:**  Review history.

---

### Acknowledgements

We would like to thank Dr. Chantriolnt-Andreas Kapourani and Dr. Alina Selega for their helpful advice with the development of diffBUM-HMM. We would like to thank Prof. Kevin Weeks for providing the Xist SHAPE-MaP mutation count data, read coverage data, and CLIPdb coordinates for the RBPs they described in the original paper. We are grateful to Dr. Krishna Choudhary and Prof. Sharon Aviran for their help with the dStruct analysis. We also thank Sergey Belikov for providing the 35S primer extension data.

### Peer review information

**Review history**

The review history is available as Additional file 2.

**Authors' contributions**

SG initiated the project. PM and GS developed the diffBUM-HMM algorithm. PM, KYTL, and SG implemented the associated software package. PM, KYTL, and SG used the diffBUM-HMM method to analyze the high-throughput RNA structure probing data for 35S pre-rRNA, mature rRNAs, and the Xist molecules, and benchmarked it against the current methods dStruct and deltaSHAPE. SG selected the data for the analyses, compared the diffBUM-HMM results to the RIP data, performed the MEME analysis on the Xist dataset, and edited the figures to conform with publication standards. PM and KYTL wrote the initial draft of the manuscript. All authors contributed to the writing of the manuscript and approved the final version.

**Authors' information**

Twitter handles: @GrannemanSander (Sander Granneman); @marangiop (Paolo Marangio).

**Funding**

This work was supported by a Medical Research Council non-Clinical Senior Research Fellowship to Dr. Sander Granneman (MR/R008205/1).

**Availability of data and materials**

All the raw and processed data files, diffBUM-HMM R code, and Python data processing pipelines used for analyzing the data in this study are available from the Granneman Lab GitLab repository (https://git.ecdf.ed.ac.uk/sgrannem/diffbum-hmm) and from Paolo Marangio's GitHub page (https://github.com/marangiop/diff_BUM_HMM) [40]. The ChemModSeq data described here are also available on the NCBI Gene Expression Omnibus (GEO) under accession numbers GSE52878 (18S rRNA data) and GSE106868 (35S pre-rRNA data). The code repository is licensed under the GNU General Public License v3.0 and is associated with the following Zenodo DOI: https://doi.org/10.5281/zenodo.4555683 [40]. PyCRAC is available from PyPI: https://pypi.org/project/pyCRAC/.

# Declarations

**Ethics approval and consent to participate**

Not applicable.

**Consent for publication**

Not applicable.

**Competing interests**

The authors declare that they have no competing interests.

**Author details**

[1]Centre for Synthetic and Systems Biology, The University of Edinburgh, Edinburgh, UK. [2]School of Informatics, The University of Edinburgh, Edinburgh, UK. [3]SISSA Data Science Excellence Department Initiative, Trieste, Italy.

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

## 
