## [**Additional file 2** Review history. · Genome Biology]

Review History

First round of review

Reviewer 1

Were you able to assess all statistics in the manuscript, including the appropriateness of statistical tests used? No.

Were you able to directly test the methods? No.

Comments to author:

This manuscript reports a novel approach for processing RNA biochemical structure probing data. The manuscript itself is very well written and easy to follow. The method reported is supported through extensive validation and testing and, once published, will add another great tool to the growing list of methods that attempt to probe RNA dynamics. This is a valuable addition to the RNA community and something that I anticipate should be useful to many. Only a few comments follow below:

Major Points:

As pointed out in the discussion (page 9, column 2, ~line 30), the resulting probabilities would be much more valuable if reported alongside differential reactivity values. The authors even suggest a solution to overcome this. It seems to be an important piece of information for potential users, since probability values alone could be misleading. From my understanding, a significant DRN could simply be an indicator of increased read depth within the region. For example, in Figure 3B, the first DRN to reach a probability of 1 (around nt 460) is on a nucleotide with a deltaSHAPE of ~0. Is this significant diffBUM-HMM DRN actually detecting a "flexibility change" between datasets? Since a relatively small change in reactivity (close to 0) could still be considered statistically significant with enough data.

In Figure 2. it is interesting that the regions with the highest/lowest deltaSHAPE values (~3800 and ~6600 respectively) are not considered diffHMM-BUM DRNs. Those are interesting areas to cover in the discussion or results, as the two method's results point to completely different conclusions.

Minor Points:

check throughout for space between number and "nt", in many cases they were together: e.g. "5nt" rather than "5 nt"

The analysis used to generate figure S2 could be described in more detail.

line 11: "10.000" should be "10000"

Line 47-8: should be "relies on counting the"

Reviewer 2

Were you able to assess all statistics in the manuscript, including the appropriateness of statistical tests used? Yes.

Were you able to directly test the methods? No.

Comments to author:

In this manuscript, Marangio et al. described DiffBUM-HMM, a method to detect differentially reactive nucleotides (DRN) from high throughput RNA structural probing data. The method is derived from BUM-HMM by expanding the hidden states from 2 (modified and unmodified) to 4 (unmodified-unmodified, unmodified-modified, modified-unmodified, and modified-modified) to accommodate differential modification status of RNA under different conditions. Benchmarking is performed against existing methods to evaluate its performance. Differential analysis on RNA structure probing assays is important. More analysis is needed to give a more quantitative and clearer picture of its performance.

Major points:

- 1) The current benchmarking is too qualitative. The authors need to provide more quantitative results for the sensitivity of DiffBUM-HMM and deltaSHAPE, particularly in the cases of pre-rRNA and Xist.
- 2) The reported 0 false positive is probably over-claiming and not convincing, given the fact that (1) in the 35S pre-rRNA analysis (Figure 1 and Figure 2), DiffBUM-HMM result is not 100% matching with PE result; and (2) the ratio between DRNs overlapping with RBP binding sites and all DRNs called by diffBUM-HMM in Xist probing assays is much lower than that of deltaSHAPE (percentage is estimated using the numbers listed in Fig 5C).
- 3) In page 3, line 17, right column, the authors mentioned "This is despite diffBUM-HMM calling 253 of the 1800 nucleotides modified in all three replicates, which is higher than the value reported for the 18S DMS datasets we previously analyzed (134)". Why the number almost doubled even though the high similarity between DiffBUM-HMM and BUM-HMM?
- 4) In the Xist part, how many of the detected DRNs are becoming more reactive from in vivo to ex vivo? The author also mentioned that from in vivo to ex vivo, the Xist might become more flexible due to loss of Ribonucleoprotein complex. So, one would expect most of the DRNs are more reactive in ex vivo condition. Does DiffBUM-HMM analysis support it?
- 5) The distribution of p-value derived from LDR/LMR distribution seems quite different from the assumed parameters for beta-distribution ($\alpha=1$, $\beta=10$) as shown in Figure S3-S5. Will it affect the convergence of EM iterations, particularly when the max iteration cycle is capped at 10 as mentioned in BUM-HMM?

Minor point:

- 1) Figure 5A, in the panel "diffBUM-HMM", what is used to plot the y-axis? Is it posterior probability or difference between posterior probability?
- 2) Figure 5D, the "den norm" is not explained only it appears in figure 6.
- 3) Figure 7A and 7B, left panels, the color in figure legend is not matching with the color used to label the DRNs.

Marangio, Law et al.: Response to Reviewers

P. Marangio, K. Y. T. Law, G. Sanguinetti, S. Granneman

23 February 2021

Overview

We thank the editor and reviewers for their time and efforts in reviewing our manuscript and for providing fair and constructive criticism. We have made the requested changes in the revised manuscript and appended detailed responses (highlighted in blue) to the individual comments along with this cover letter. The changes that we made in the main text are highlighted in yellow.

Below we summarize the major changes incorporated:

- We now present the manuscript using BioMed Central’s TeX template.
- Added new supplementary figure in manuscript (S6) plotting LDR differences alongside output of diffBUM-HMM, as an indicator of differential reactivity
- Added new supplementary figure in manuscript (S7) that shows deltaSHAPE results for the two 35S ChemModSeq biological replicates to illustrate the variability in signal in two regions.
- We included results from diffBUM-HMM analyses performed on additional SHAPE-Seq control datasets, which reinforces our conclusion that diffBUM-HMM calls few false positives.
- Ensured manuscript conforms to standard of “Methods” articles of *Genome Biology*
- Added a substantial amount of clarifications inside the manuscript text and captions, with close regard to the reviewers’ comments
- Restructured the Discussion section by adding subheadings in order to separate topics of discussion, and greatly expanded the discussion of the interpretation of diffBUM-HMM outputs.

Aside from the suggested revisions, we have repeated the dStruct analyses with a new set of parameters (search window of 5 nucleotides and false discovery rate (FDR) set to 0.15). These were chosen based on recommendations from the first author of the dStruct manuscript, with whom we shared our results before submitting to Genome Biology, our preference for higher specificity of detected sites and our desire to do an accurate and fair comparison between diffBUM-HMM and dStruct. Figures 2, 3 and 5A have been updated with the new results, and are described in lines 201-226 and 340-352 of the manuscript. While the new search length increased the number of DRRs reported by dStruct, many of these are reported with FDRs higher than 0.05. Thus, although reducing the search window from 11 to 5 increases the number of DRRs called by dStruct, it also decreased the confidence in the DRRs it reported. Additionally, dStruct still remains very conservative in comparison to the number of DRNs detected by diffBUM-HMM. The DRRs detected by dStruct are also quite extensive, and it is not possible to pinpoint those nucleotides that are differentially reactive. Thus, changing the dStruct parameters has not changed our conclusion that diffBUM-HMM has a much better sensitivity and, unlike dStruct, has single nucleotide resolution.

Finally, we are also pleased to report that our work has already had significant impact. The diffBUM-HMM algorithm was used in a manuscript published in *The Plant Cell* in January 2021 (Reis et al. 2021; <https://doi.org/10.1093/plcell/koab010>) to study the structural changes in a plant mRNA.

Reviewer #1

This manuscript reports a novel approach for processing RNA biochemical structure probing data. The manuscript itself is very well written and easy to follow. The method reported is supported through extensive validation and testing and, once published, will add another great tool to the growing list of methods that attempt to probe RNA dynamics. This is a valuable addition to the RNA community and something that I anticipate should be useful to many. Only a few comments follow below:

We thank the reviewer for the positive feedback and constructive criticism.

1) As pointed out in the discussion (page 9, column 2, line 30), the resulting probabilities would be much more valuable if reported alongside differential reactivity values. The authors even suggest a solution to overcome this. It seems to be an important piece of information for potential users, since probability values alone could be misleading.

We thank the reviewer for highlighting this issue, which we agree is important and warrants further discussion. We followed the reviewer's suggestion and have

made a new Supplementary Figure 6 where we plotted the scaled drop-off rates (DORs) that diffBUM-HMM produces for each condition alongside the posterior probabilities. As discussed in the revised manuscript, the scaled DORs can provide a good indication of the nucleotides reactivity to chemical probes. This is summarized in lines 575-582. Additionally, we have expanded the relevant discussion in lines 583-621, and added a table of P values in Supplementary Figure S6B, to provide readers with further information on interpreting the probabilistic output, illustrated with examples from our 35S pre-rRNA dataset.

2) From my understanding, a significant DRN could simply be an indicator of increased read depth within the region. For example, in Figure 3B, the first DRN to reach a probability of 1 (around nt 460) is on a nucleotide with a deltaSHAPE of 0. Is this significant diffBUM-HMM DRN actually detecting a "flexibility change" between datasets? Since a relatively small change in reactivity (close to 0) could still be considered statistically significant with enough data.

Indeed the observations and conceptual understanding of the reviewer are accurate. The probabilistic output of diffBUM-HMM is meant to pinpoint regions where the proportion of differentially reactive molecules is sufficiently large between the conditions, and cannot be explained by random variability alone. In the case of position 462 in the 5'ETS, although the differences in SHAPE reactivity are modest between the wild-type Mrd1 and the mutant, diffBUM-HMM finds that these differences were reproducibly detected between the samples and therefore diffBUM-HMM called the position more reactive in the wild-type Mrd1 data. If the user is specifically interested in studying large changes in chemical reactivities, they could use the posterior probabilities as a way to filter out the differentially reactive nucleotides and use the scaled DORs to identify regions with large changes.

As for the possibility of increased read depth biasing small changes in reactivity towards significant DRN calls, a few aspects of the diffBUM-HMM design accounts for artifacts in the data that could be introduced in the library preparation procedure. These are shared by its predecessor BUM-HMM. Coverage biases are addressed with the `stabilizeVariance` function built in the data pre-processing steps of the pipeline (see: <http://bioconductor.org/packages/release/bioc/html/BUMHMM.html>). Log-drop-off rates or log-mutation rates (LDRs/LMRs), which quantify differences in nucleotide reactivity between treated and control samples, are transformed to remove their dependence on coverage. Hence, increased read depth within the region is not likely to contribute to this discrepancy between diffBUM-HMM and deltaSHAPE at the particular nucleotide position pointed out by the reviewer.

Each diffBUM-HMM analysis also takes into consideration inter-replicate variability, whereas deltaSHAPE only represents the differential reactivity detected between a single pair of differentially probed samples. Essentially the prediction

of the diffBUM-HMM DRNs is based on a more comprehensive set of information, than a typical deltaSHAPE analysis, so it could be picking up reactivity differences that this particular deltaSHAPE analysis does not. Sample variability and background signals are also accounted for very differently between the two methods, so discrepancies between diffBUM-HMM and deltaSHAPE are possible. We have now added this remark to the discussion at lines 622-650, under the section "Interpreting the output of diffBUM-HMM".

We hope that the new explanations adequately address the reviewer's concerns.

3) In Figure 2. it is interesting that the regions with the highest/lowest deltaSHAPE values (~3800 and ~6600 respectively) are not considered diffHMM-BUM DRNs. Those are interesting areas to cover in the discussion or results, as the two method's results point to completely different conclusions.

We agree that this requires more clarification. We have two sets of deltaSHAPE values since there are two biological replicates for each variant, and the peaks observed at around nt 3800 and nt 6600 are peaks that appear only in one of the two replicates. DiffBUM-HMM accounts for inter-replicate variability and background noise by including data from experimental replicates and untreated samples, On the other hand, deltaSHAPE does not account for inter-replicate variability. Hence, the discrepancy that we see between deltaSHAPE and diffBUM-HMM in the aforementioned two regions, is because the variability between the two treated replicates was too large, such that they were regarded as noisy signals. Therefore diffBUM-HMM did not call them as differentially modified. You can see clear examples of that in the Xist data shown in Figure 7, where we show the deltaSHAPE results for both individual replicates as well as the diffBUM-HMM results. There are several regions where deltaSHAPE only calls differentially modified nucleotides in only one replicate and in many of those cases did diffBUM-HMM give these nucleotides low posterior probabilities of differential modification. Additionally, we now discuss these discrepancies in the Discussion section in more detail, at lines 622-650, and we have included a new Supplementary Figure 7 in the revised manuscript that shows the variability in the deltaSHAPE results for two regions in the 35S pre-rRNA ChemModSeq data.

4) Check throughout for space between number and "nt", in many cases they were together: e.g. "5nt" rather than "5 nt"
This has been amended in the main text, in figures and in captions throughout the resubmitted manuscript.

5) The analysis used to generate figure S2 could be described in more detail. We have expanded the legend of Figure S2 and included more information about how these data were generated. We now also include the names of the proteins that we predict in these sequences next to each motif. For more details on the

MEME analyses, we refer to the "Analysis of enriched sequence motifs in regions containing DRNs" subsection under the Methods section.

6) line 11: "10.000" should be "10000"

We have made the requested change.

7) Line 47-8: should be "relies on counting the"

We have made the requested change.

Reviewer #2

In this manuscript, Marangio et al. described DiffBUM-HMM, a method to detect differentially reactive nucleotides (DRN) from high throughput RNA structural probing data. The method is derived from BUM-HMM by expanding the hidden states from 2 (modified and unmodified) to 4 (unmodified-unmodified, unmodified-modified, modified-unmodified, and modified-modified) to accommodate differential modification status of RNA under different conditions. Benchmarking is performed against existing methods to evaluate its performance. Differential analysis on RNA structure probing assays is important. More analysis is needed to give a more quantitative and clearer picture of its performance.

We thank the reviewer for providing positive and constructive feedback.

Major Points: 1) The current benchmarking is too qualitative. The authors need to provide more quantitative results for the sensitivity of DiffBUM-HMM and deltaSHAPE, particularly in the cases of pre-rRNA and Xist.

We completely agree with the reviewer that having more quantitative results would be beneficial and would provide a more rigorous benchmarking standard. However, we simply do not have an absolute "ground truth" for the structural changes that the RNA molecules undergo in the differential conditions we assessed. To generate such a ground truth we would need high-resolution X-ray or cryo-EM structures of these molecules generated under the same conditions so that we test how well our diffBUM-HMM data agrees with the structural data and compared it to results from other tools. Unfortunately, we do not have high resolution structures for the complexes that we analysed. We have also looked at alternative datasets that we could use. However, our pipeline also requires RNA probing datasets that were generated under two different conditions using NGS technologies and have biological replicates for treated and control samples. Such datasets are very limited in availability. The authors of the dStruct paper (Choudhary et al.; <https://doi.org/10.1186/s13059-019-1641-3>) had a similar problem. They eventually analysed SHAPE-seq data generated by the Lucks lab to demonstrate that dStruct can reliably detect structural changes in

a bacterial riboswitch. However, this RNA is short (about 100 nt) and contains only four regions that are known to undergo structural changes in the presence of a ligand. This is not sufficient for doing statistically meaningful quantitative analyses. Ideally, we would like to have high-throughput RNA structure probing datasets and high-resolution structural data of a large complex, such as a ribosome, in two different states.

Nonetheless, the reviewer raises an important point and therefore we now discuss this limitation in more detail in the Discussion section of our revised manuscript, in lines 518-543.

2) The reported 0 false positive is probably over-claiming and not convincing, given the fact that (1) in the 35S pre-rRNA analysis (Figure 1 and Figure 2), DiffBUM-HMM result is not 100% matching with PE result; and (2) the ratio between DRNs overlapping with RBP binding sites and all DRNs called by diffBUM-HMM in Xist probing assays is much lower than that of deltaSHAPE (percentage is estimated using the numbers listed in Fig 5C).

In order to make the analysis of false positive calls more robust, we analysed another 18S rRNA DMS probing dataset that we had previously used to benchmark the BUM-HMM algorithm (Selega et al.; <https://www.nature.com/articles/nmeth.4068>). This dataset contains two biological replicates of untreated and DMS treated samples. We analysed this dataset using diffBUM-HMM, dStruct and deltaSHAPE. We now show these results in Table 1 and Figure 4 (18S #2 dataset). Remarkably, we obtained identical results for diffBUM-HMM with this dataset. Zero false-positives were reported by diffBUM-HMM and dStruct, whereas deltaSHAPE reported 16.

We agree that the diffBUM-HMM results are not 100% in agreement with the PE results, however, it is important to point out that this problem is not because diffBUM-HMM calls false positives, but because there are some differences between the ChemModSeq data and the PE data. Note that diffBUM-HMM does not consider the PE data when calling differentially modified nucleotides so some differences between the difBUM-HMM and PE results are to be expected.

With respect to the Xist data, there are a number of reasons why we think that the enrichment of the RBPs that we analysed is lower in the diffBUM-HMM results compared to deltaSHAPE: Firstly, Xist is associated with a large number of proteins, for many of which we do not have RNA-binding datasets available. It is possible that a lot of the structural changes we see are linked to other RBPs. Although our Xist analysis do not allow us to completely exclude the possibility that diffBUM-HMM calls more false-positives in Xist RNA than deltaSHAPE, we think this is unlikely for the following reasons. Firstly, if many of these calls on the Xist data were indeed false positives, then we would expect that they would be randomly distributed throughout the RNA. This is not the case as we found that more than 80% of the nucleotides that were uniquely

called differentially modified by diffBUM-HMM were located in single-stranded regions. In addition, we would not necessarily expect to see an enrichment of A's and U's in this group.

3) In page 3, line 17, right column, the authors mentioned "This is despite diffBUM-HMM calling 253 of the 1800 nucleotides modified in all three replicates, which is higher than the value reported for the 18S DMS datasets we previously analyzed (134)". Why the number almost doubled even though the high similarity between DiffBUM-HMM and BUM-HMM?

We agree that this sentence is ambiguous, so we have decided to remove it from the manuscript. What we intended to convey was that although diffBUM-HMM did not report any false positives on the 18S DMS Structure-Seq data (generated by the Aviran Lab), we could not exclude the possibility that this dataset was not of a good quality. It is entirely possible that the variability between the replicates was so high that diffBUM-HMM is not even able to call nucleotides that were modified in all replicates. The fact that diffBUM-HMM called 253 modified nucleotides in this dataset strongly suggests that the data is of good quality as otherwise the tool would not call so many positions. Previously, we used BUM-HMM to analyse a DMS 18S rRNA dataset that we generated a few years ago in our own lab (Selega et al.; <https://www.nature.com/articles/nmeth.4068>). For this dataset BUM-HMM called 134 modified nucleotides, so we were expecting to see a number in that range. In fact, we now also analysed this dataset with diffBUM-HMM (Table 1; 18S #2 dataset) and found that the tool calls 0 false-positives on this dataset, and 132 nucleotides being modified in both samples.

4) In the Xist part, how many of the detected DRNs are becoming more reactive from in vivo to ex vivo? The author also mentioned that from in vivo to ex vivo, the Xist might become more flexible due to loss of Ribonucleoprotein complex. So, one would expect most of the DRNs are more reactive in ex vivo condition. Does DiffBUM-HMM analysis support it?

Yes, the diffBUM-HMM data indeed support the idea that the RNA becomes more flexible in the *ex vivo* condition. We apologise for not making this clear enough in the main text. We have now changed the text in the paragraph describing these results to make this more evident in lines 326-340, quoted below:

Interestingly, diffBUM-HMM reported a much larger number of DRNs in the ex vivo data than in the in vivo data relative to deltaSHAPE (≈ 9 -fold with diffBUM-HMM and ≈ 1.4 -fold with deltaSHAPE, as shown in Figs. 5A and C). Why diffBUM-HMM calls many more DRNs in the ex vivo data is unclear, however, intuitively one would expect that removing proteins from a very large ribonucleoprotein (RNP) complex will substantially increase in the flexibility of the RNA. This is because when the RNP complex is deproteinized, those nu-

cleotide positions normally bound by RBPs may become more accessible and/or more flexible and therefore react more readily with chemical probes.

5) The distribution of p-value derived from LDR/LMR distribution seems quite different from the assumed parameters for beta-distribution (alpha=1, beta=10) as shown in Figure S3-S5. Will it affect the convergence of EM iterations, particularly when the max iteration cycle is capped at 10 as mentioned in BUM-HMM?

This an important point. We discuss this in detail in the revised manuscript in the "DiffBUM-HMM model" subsection under the Methods section, where we explain why there may be a difference between the observed LDR/LMR distributions and the assumed model parameters. Additionally, we have expanded this subsection to address the reviewer's question about the convergence of the EM algorithm, in lines 758-766: Running the EM algorithm is an optional step in the diffBUM-HMM pipeline, however, we did not perform this step for any of the experiments presented in the manuscript as it was not necessary. We hope this clarifies this aspect.

6) Figure 5A, in the panel "diffBUM-HMM", what is used to plot the y-axis? Is it posterior probability or difference between posterior probability?

The y-axis is posterior probability of differential modification. The positive y axis indicates positions that are more reactive under *ex vivo* conditions (are more accessible to chemical modification when *ex vivo*/were protected from modification when *in vivo*), while those with enhanced reactivity when *in vivo* are shown on the negative y axis. We have now made this clearer in the figure legend.

7) Figure 5D, the "den norm" is not explained only it appears in figure 6.

We have added additional text in the caption of Figure 5D explaining what "normalized" and "den norm" refers to.

8) Figure 7A and 7B, left panels, the color in figure legend is not matching with the color used to label the DRNs.

We apologise for the error. This has been amended accordingly in the revised manuscript.

Comments from the Editor

1) When revising the manuscript, please ensure the manuscript conforms to our style for Methods articles (see <https://genomebiology.biomedcentral.com/submission-guidelines/preparing-your-manuscript/method>); specifically,

the abstract should be under 100 words.

We have reduced the size of the abstract to 100 words, and implemented all the required changes to conform to the style of *Genome Biology* for Methods articles.

2) Please note that if we decide to publish your manuscript we will require that the source code is made publicly available under an open source license compliant with Open Source Initiative, with the license clearly stated in the manuscript. The source code should be deposited in a public repository, such as for instance github, with the accession links included in the manuscript. We also ask that the version of source code used in the manuscript is deposited in a DOI-assigning repository, such as zenodo, with the link also included. All this information should be listed in a separate Availability of Data and Materials section of the manuscript.

A link to a GitHub page with the source code is given in line 816 in the subsection "Availability of data and materials" under "Methods" section. We also added a sentence at lines 819-820 clarifying the license of the source code as well. The license is also clearly stated in the source code in Github. The source code of the Github repository has been uploaded to Zenodo and is associated with the following DOI: <https://doi.org/10.5281/zenodo.4555683>. The Zenodo DOI has been included in the manuscript at line 821 in the manuscript.

Second round of review

Reviewer 1

The authors have addressed all my comments and concerns.